# Influence of the Epoxy/Acid Stoichiometry on the Cure Behavior and Mechanical Properties of Epoxy Vitrimers

**DOI:** 10.3390/molecules27196335

**Published:** 2022-09-26

**Authors:** Fan Jing, Ruikang Zhao, Chenxuan Li, Zhonghua Xi, Qingjun Wang, Hongfeng Xie

**Affiliations:** 1MOE Key Laboratory of High Performance Polymer Materials and Technology, School of Chemistry and Chemical Engineering, Nanjing University, Nanjing 210093, China; 2Department of Chemistry, Texas A&M University, College Station, TX 77843, USA; 3Experimental Chemistry Teaching Center, School of Chemistry and Chemical Engineering, Nanjing University, Nanjing 210023, China

**Keywords:** epoxy vitrimer, stoichiometry, cure behavior, solvent stability, glass transition temperature, mechanical properties

## Abstract

Bisphenol A epoxy resin cured with a mixture of dimerized and trimerized fatty acids is the first epoxy vitrimer and has been extensively studied. However, the cure behavior and thermal and mechanical properties of this epoxy vitrimer depend on the epoxy/acid stoichiometry. To address these issues, epoxy vitrimers with three epoxy/acid stoichiometries (9:11, 1:1 and 11:9) were prepared and recycled four times. Differential scanning calorimetry (DSC) was used to study the cure behavior of the original epoxy vitrimers. The dynamic mechanical properties and mechanical performance of the original and recycled epoxy vitrimers were investigated by using dynamic mechanical analysis (DMA) and a universal testing machine. Furthermore, the reaction mechanism of epoxy vitrimer with different epoxy/acid stoichiometry was interpreted. With an increase in the epoxy/acid ratio, the reaction rate, swelling ratio, glass transition temperature and mechanical properties of the original epoxy vitrimers decreased, whereas the gel content increased. The recycling decreased the swelling ratio and elongation at break of the original epoxy vitrimers. Moreover, the elongation at break of the recycled epoxy vitrimers decreased with the epoxy/acid ratio at the same recycling time. However, the gel content, tensile strength and toughness of the original epoxy vitrimers increased after the recycling. The mechanical properties of epoxy vitrimers can be tuned with the variation in the epoxy/acid stoichiometry.

## 1. Introduction

Epoxy resin, one of the most important and popular classes of thermosetting polymers, is firmly rooted in many areas, both in our daily lives and in industry [1,2,3,4,5]. Due to its outstanding mechanical strength, dimensional and thermal stability, as well as creep, chemical and electrical insulation resistance, epoxy resin has been widely used in construction, adhesives, electronic and electrical devices, coatings, composites and so on [1,6,7,8,9,10]. However, due to their insoluble and infusible nature, the irreversible covalent networks restrict epoxy resin from being recycled and reprocessed. Thus, most of the epoxy wastes are disposed of by landfill or incineration, which has caused not only a waste of resources but also environmental pollution. To solve these problems, a thrust to develop a novel class of thermosetting polymers with recyclable, healable and reprocessable features has been carried out during the last two decades [11,12].

In 2011, Leibler et al. developed a new class of polymers, called vitrimer, based on associative covalent adaptable networks of epoxy resins cured with a mixture of dimerized and trimerized fatty acids through transesterification reactions [13]. Vitrimers contain crosslinked networks constituted by dynamic covalent bonds, which can undergo exchangeable reactions without changing the crosslink density. In this case, vitrimers behave like traditional thermosetting polymers, with good mechanical and thermal properties at service temperature. However, exchangeable reactions happen when heated to higher temperatures over the topology freezing transition temperature (T_v_), resulting in a topology rearrangement under external force. Under this circumstance, vitrimers can flow like vitreous fluid [14,15]. The exchangeable networks enable epoxy vitrimers to be recycled, reshaped and reprocessed at a high temperature [16,17,18].

Epoxy resins cured with carboxylic acids and anhydrides are the most studied epoxy vitrimers [13,19,20]. During fabrication, epoxy monomers or oligomers containing epoxide groups react with acids/anhydrides at a stoichiometry of 1:1 (epoxy:carboxyl) molar ratio under the acceleration of transesterification catalysts, which generates numerous ester bonds and hydroxyl groups. As a result, transesterification between ester bonds and hydroxyl groups takes place at elevated temperatures, which induces a topology network arrangement and, thus, makes the epoxy vitrimers malleable [12]. The properties of epoxy vitrimers are directly influenced by the network structure, which is strongly dependent on the chemical structure of the curing agent and the epoxide equivalent of epoxy oligomers [14,21,22]. Thus, to tune both mechanical and thermal properties of epoxy vitrimers, starting materials with an off-stoichiometric ratio, especially epoxy-to-carboxyl ratio > 1, were used [23,24,25]. It is well understood that excess epoxy is propitious to provide enough free hydroxyl groups for transesterification. For industrial epoxy/anhydride systems, the stoichiometric ratio of epoxy to anhydride is from 1:0.8 to 1:0.9 [26]. However, the cured epoxy networks cannot be recycled at such stoichiometric ratios since the amount of free hydroxyl groups is too low for transesterification [27].

The long-chain dicarboxylic fatty acid can be obtained through the dimerization of fatty acids. Because of side reactions, the dimerized fatty acid is a mixture of monomers, dimers and trimers [28]. Since 2011, much attention has been paid to vitrimers cured with a mixture of dimerized and trimerized fatty acids in the presence of transesterification [13,14,15,29,30,31,32]. Yu and coworkers studied the effect of the stoichiometry of epoxy and fatty acid/anhydride on the glass transition temperature (T_g_) and bond exchange reactions of epoxy vitrimers [33]. Both the T_g_ and energy barrier of the vitrimer increase with the anhydride content in fatty acid and anhydride mixtures. Snijkers and coworkers compared the curing of epoxy vitrimers cured with three fatty acid mixtures [34]. The activation energy and gelation time of epoxy vitrimers increase with the average functionality of the fatty acid mixture. It is worth mentioning that the stoichiometry of epoxy and carboxyl in most epoxy vitrimers cured with a mixture of fatty acids is fixed at 1:1. Although the cure behavior and properties of epoxy resin with different stoichiometries of epoxy oligomers and curing agents have been studied [35], the impact of off-stoichiometry of epoxy/acid on the mechanical properties and thermal properties of epoxy vitrimers is yet to be investigated.

In this work, the effect of the stoichiometry of epoxy and acid on vitrimer properties was investigated. Three epoxy vitrimers were prepared, using a commercial epoxy oligomer, a mixture of fatty acids with three stoichiometries (from 9:11 to 11:9 epoxy/acid ratio) as the curing agent and zinc acetate (Zn(Ac)_2_) as the catalyst. The cure behavior of epoxy vitrimers was determined by differential scanning calorimetry (DSC). To investigate the effect of recycling on the solvent stability, thermal and mechanical properties of the original epoxy vitrimers, solvent immersion, dynamic mechanical analysis (DMA) and uniaxial tests were performed.

## 2. Materials and Methods

### 2.1. Materials

Bisphenol A epoxy oligomer (trade name of 0164) was obtained from Nantong Xingchen Synthetic Material Co., Ltd. (Nantong, China). A mixed fatty acid with about 23 wt% dimerized acid and 77 wt% trimerized acid was supplied by Shanghai Zhipu Chemical Co., Ltd. (Shanghai, China). Zinc acetate (Zn(Ac)_2_) was purchased from Shanghai Macklin Biochemical Co., Ltd. (Shanghai, China). Table 1 lists detailed information about molecular weights and chemical structures of each reagent for epoxy vitrimers.

### 2.2. Preparation of Epoxy Vitrimer

Zinc acetate was mixed with the mixture of fatty acids to prepare a masterbatch of curing agents in a 500 mL beaker at 900 rpm at 180 °C for 2 h. The content of Zn(Ac)_2_ was 10 mol% of the COOH groups. The epoxy oligomer was introduced into the masterbatch and stirred at 1500 rpm for 3 min at 120 °C. Finally, the mixture was quickly poured into a PTFE (polytetrafluoroethylene) mold with a diameter of 100 mm and a height of 5 mm and placed in an oven at 120 °C for 4 h. Figure 1 presents a schematic illustration for the preparation of epoxy vitrimer. The stoichiometries of the epoxy to acid of epoxy vitrimers were 9:11, 1:1 and 11:9 and the respective samples were named as EV45, EV50 and EV50.

### 2.3. Recycling of Epoxy Vitrimers

Cured epoxy vitrimer film was cut into small pieces and then reprocessed to bulk material by the hot press at 180 °C and 10 MPa for 20 min. Then, the recycled sample was cut into small pieces and reprocessed again. The recycling process was repeated three times at a 10 °C increase in temperature each time. The reprocessed samples for the first, second, third and fourth times were abbreviated as 1R, 2R, 3R and 4R, respectively. Figure 2 represents a schematic illustration for the preparation and recycling of epoxy vitrimer.

### 2.4. Methods

Tensile tests were conducted on an Instron 3366 testing machine according to ASTM D638 (type V). The dumbbell-shaped samples were tested at room temperature. The crosshead speed during measurements was 200 mm/min. At least five samples were measured for each composition. DMA was carried out on a DMA + 450 dynamic mechanical analyzer (01 dB Metravib, France). The samples with dimensions of 30 × 20 × 2.5 mm^3^ were mounted on a tension clamp and tested at a ramping rate of 3 °C/min and a frequency of 1 Hz from −25 to 125 °C. The T_g_ was determined as the temperature of the maximum damping factor (tan δ) value in a tan δ versus temperature curve. Isothermal and dynamic DSC analyses were performed on a DSC 1 instrument (Mettler-Toledo, Switzerland) under a nitrogen flow of 20 mL/min. For isothermal curing, the uncured sample (~20 mg) was heated rapidly to 120 °C and kept at that temperature until the heat flow leveled off to the baseline. The isothermal heat of reaction (Δ*H*_i_) was calculated by the integration of the exothermal curve in time. After the isothermal scan, the sample was cooled rapidly to 20 °C and then reheated to 300 °C at a heating rate of 10 °C/min to determine the heat of reaction during dynamic curing (Δ*H*_t_). The swelling ratio (*SR*) and gel content (*GC*) of cured networks were tested in toluene at room temperature for 48 h. After wiping the solvent, the swollen sample was weighed and dried at 80 °C for 24 h in a ventilated oven to remove the toluene. The *SR* and *GC* were calculated as follows:(1)$$SR=\frac{w_{s}-w_{0}}{w_{0}}$$
(2)$$GC=\frac{w_{d}}{w_{0}}$$
where $w_{s}$ is the weight of the swollen sample, $w_{0}$ is the initial weight and $w_{d}$ is the weight of the dried sample.

## 3. Results and Discussion

### 3.1. Cure Behavior

DSC is a powerful tool for monitoring the cure reaction of epoxy resins [36,37]. To determine the cure behavior of epoxy vitrimers, the uncured sample was cured by isothermal curing at 120 °C followed by dynamic curing. As shown in Figure 3, all epoxy vitrimers exhibit an exothermic peak during the isothermal curing and dynamic curing. The peak area of dynamic curing increases with the epoxy/acid ratio. The cure reaction of epoxy resins can be divided into two distinct stages: chemical control and diffusion control [38]. Chemical control reaction occurs at the beginning of curing and dominates until the appearance of vitrification, at which the cure reaction becomes very slow and finally stops, known as a diffusion-controlled reaction. In this circumstance, the heat flow levels out during the isothermal curing (Figure 3a). However, during the subsequent dynamic curing, the higher temperature results in the continuing reaction of epoxy. Thus, an exothermic peak appears (Figure 3b).

It is believed that the instantaneous reaction rate (*dα*/*dt*) is proportional to the heat flow (*dH*/*dt*) during a cure reaction
(3)$$\frac{d\alpha }{dt}=\frac{dH/dt}{\Delta H_{i}+\Delta H_{d}}$$
where *α* is the conversion (extent of reaction) and $\Delta H_{i}+\Delta H_{d}$ is the total heat generated during isothermal and dynamic curings. The conversion is given by
(4)$$\alpha =\frac{\Delta H_{t}}{\Delta H_{i}+\Delta H_{d}}$$
where $\Delta H_{t}$ is the heat generated at a certain time in an isothermal DSC run.

As shown in Figure 3a, all epoxy vitrimers exhibit an autocatalytic reaction during isothermal curing. Moreover, the heat generated at the dynamic curing increases with the epoxy/acid ratio (Figure 3b). The conversion of epoxy vitrimers begins to decrease with the epoxy/acid ratio after 10 min curing at 120 °C, as shown in Figure 3c. When curing at 120 °C for 50 min, the conversions of EV45, EV50 and EV55 are 0.76, 0.72 and 0.61, respectively, indicating that the conversion and reaction rate of the epoxy vitrimers decreases with the epoxy/acid ratio.

The cure reactions of epoxides with acids in epoxy vitrimers are presented in Figure 4. Five main reactions are considered [21,25,39,40]: the polyaddition of epoxides and acids, forming the characteristic β-hydroxyl ester of epoxy vitrimer (1), ring-opening polymerization (ROP) via hydroxyl groups (2), condensation–esterification of acids and hydroxyl groups (3), catalytic ROP between epoxides, forming ether bonds (4) and transesterification of β-hydroxyl ester (5). For the epoxy vitrimer with a 1:1 epoxy/acid ratio, main Reactions (1, 2 and 5) take place. In addition, Reactions (2 and 3) or Reaction (4) occurs with the excess of acids or epoxides. It is worth noting that Reaction (4) generally takes place at an elevated temperature [41]. In addition, the steric hindrance of long-chain curing agents limits the reaction of epoxy resins [42,43]. Therefore, EV55 exhibits the lowest reaction rate and conversion during the isothermal curing among all epoxy vitrimers, as shown in Figure 3. For EV45, Reaction (2) is more pronounced than EV50 due to the excess of acid and the existence of the catalyst. In this case, EV45 exhibits the highest reaction rate and conversion among all epoxy vitrimers.

### 3.2. Solvent Stability and Gel Content

To determine the solvent stability of the original and recycled epoxy vitrimers, samples were immersed in toluene at room temperature for 48 h. As shown in Figure 5, except for EV55, all samples are stable in toluene after being immersed in toluene for 48 h.

Figure 6 depicts the swelling ratios and gel contents of the original and recycled epoxy vitrimers. For the original epoxy vitrimers, with the increase in the epoxy/acid ratio, the swelling ratio increases and the gel content decreases, indicating that the crosslinking density of epoxy vitrimers decreases with the epoxy/acid ratio. This trend is opposite to the vitrimer system of tetrafunctional epoxy/dimerized acid [44]. After recycling, the swelling ratio of the original epoxy vitrimers, especially EV55, lowers, as shown in Figure 6a. For recycled epoxy vitrimers, the swelling ratio of the recycled epoxy vitrimers decreases with the recycling time. In addition, the swelling ratio of recycled EV55 is higher than those of recycled EV50 and EV45 at the same recycling time. In contrast to the swelling ratio, the recycling increases the gel content of the original epoxy vitrimers, especially for EV55. For EV50 and EV55, the gel content increases with the recycling time. However, the recycling time has little effect on the gel content of EV45 due to their high crosslinking density. It is important to mention that the highest swelling (146%) and lowest gel content (71%) of EV55 are caused by the lowest reaction rate and occurrence of catalytic ROP between epoxides at an elevated temperature, as mentioned previously. In this case, some sol of EV55 is soluble in toluene (Figure 5) due to its lowest crosslinking density. However, after recycling at a higher temperature, catalytic ROP continues. Thus, the swelling ratio decreases and the gel content increases with the recycling time. Furthermore, the increased crosslinking density improves the solvent stability of recycled EV55.

### 3.3. Dynamic Mechanical Properties

#### 3.3.1. Storage Modulus (E′)

Figure 7 illustrates storage modulus–temperature curves of the original and recycled epoxy vitrimers. The effect of recycling on the E′ of the original epoxy vitrimers depends on the epoxy/acid ratio. In the glassy state, the E′ of the recycled EV45 is higher than that of the original one. However, during the glass transition, an opposite trend appears. In the rubbery state, the E′ of the recycled EV45 is not lower than that of the original one. For EV50, the effect of recycling on the original vitrimer is complicated. After the fourth recycling, the E’ of the recycled vitrimer is higher than that of the original one within the whole temperature interval. Due to the significantly increased crosslinking density after the high-temperature recycling, all recycled vitrimers of EV55 have a higher E′ than the original one. In addition, the E′ value increases with the recycling time during and after the glass transition.

#### 3.3.2. Loss modulus (E″)

Figure 8 shows loss modulus–temperature curves of the original and recycled epoxy vitrimers. Loss modulus indicates the viscous response of the viscoelastic material and describes the molecular motions of the polymers [45]. All loss modulus–temperature curves exhibit a glass transition peak. For EV45, the glass transition peak shifts to a lower temperature after recycling (Figure 8a). However, EV55 shows an opposite trend (Figure 8c). Before the glass transition, the loss moduli of EV45 and EV55 are lower than those of recycled ones. However, the loss modulus of EV45 is higher than those of recycled ones during the glass transition and at the rubbery state (Figure 8a). For EV55, the trend of loss modulus during the glass transition and at the rubbery state is the same as the one before the glass transition. The effect of recycling on the loss modulus of EV50 is complicated. Before the glass transition, the loss modulus of EV50 is the same as that of the fourth recycling, but is higher than those of first, second and third recycling (Figure 8b). During the glass transition and at the rubbery state, the loss modulus of EV50 is the same as that of the third recycling, higher than that of the first recycling and lower than those of the second and fourth recycling.

#### 3.3.3. Glass Transition Temperature

Figure 9 presents the damping factor (tan δ)–temperature curves of the original and recycled epoxy vitrimers. In a tan δ versus temperature curve, the peak indicates the occurrence of glass transition. The temperature at the maximum tan δ is often defined as the glass transition temperature [46,47,48]. The tan δ–temperature curves of EV50 and EV55 exhibit only one tan δ peak at ~25 °C, indicating that these vitrimers have a single glass transition from glassy to rubbery state, also known as major transition or α transition due to the gradual chain movements. Apart from the similar glass transition to EV50 and EV55 at ~25 °C, it is interesting to note that another broad peak appears between 40 °C and 100 °C in the tan δ–temperature curve of EV45. In a polymer blend, this shoulder indicates another glass transition and two distinct glass transitions indicate the immiscibility between two components [49,50]. As shown in Figure 8a, no β glass transition peak appears after the α transition. Therefore, the original and recycled EV45 have another α transition, which may be attributed to the ring-opening polymerization via hydroxyl groups (Reaction 2), as discussed in Section 3.1.

Figure 10 depicts the T_g_s of the original and recycled epoxy vitrimers. The T_g_ of the original epoxy vitrimers decreases with the epoxy/acid ratio. A similar trend was reported in the tetrafunctional epoxy/dimerized acid vitrimer [44]. The variation in the T_g_ with the epoxy/acid ratio can be attributed to the crosslink density of epoxy vitrimers [33]. As discussed in Section 3.1, EV45 exhibits the fastest reaction rate, resulting in the highest crosslink density. On the contrary, the slowest reaction rate of EV55 leads to the lowest crosslink density. In this case, the T_g_ of the original epoxy vitrimers decreases with the epoxy/acid ratio.

For EV45, the recycling decreases the T_g_ of the original vitrimer. In addition, except for the fourth recycling, the T_g_ of recycled vitrimers slightly decreases with the recycling time. However, EV55 shows an opposite trend since the occurrence of the catalytic ring-opening polymerization between epoxides occurs at a high recycling temperature, resulting in an increase in the crosslink density of the original epoxy vitrimer. For the recycled EV55, the T_g_ increases in the recycling time since the gradual increase in the recycling temperature brings about more catalytic ring-opening polymerization and, thus, a further increase in the crosslink density.

### 3.4. Mechanical Properties

Figure 11 depicts the stress–strain curves of the original and recycled epoxy vitrimers obtained from the tensile tests. All samples behave as ductile materials with high strains but low stresses. In stress–strain curves, tensile strength is the maximum stress that a material can withstand before breaking while the stain at break is also called elongation at break [51].

Figure 12 presents the tensile strength of the original and recycled epoxy vitrimers. The tensile strength of the original vitrimers decreases with the epoxy/acid ratio. However, a contrary trend was shown in the tetrafunctional epoxy/dimerized acid vitrimers [44]. It is known that both thermal and mechanical properties of epoxy resins depend on crosslink density [52]. Higher crosslink density indicates both higher T_g_ and tensile strength. Therefore, the decrease in the tensile strength is attributed to the decrease in the crosslink density of the original epoxy vitrimers with the epoxy/acid ratio. The recycling significantly increases the tensile strength of the original vitrimers except for the first recycling. For the tetrafunctional epoxy/dimerized acid vitrimers, an opposite trend was shown [44]. For the recycled epoxy vitrimers, the tensile strength exhibits the same trend as the original epoxy vitrimers after the first and second recycling. However, the tensile strength of the recycled epoxy vitrimers declines in the following sequence: EV50 > EV55 > EV45 after the third and fourth recycling.

Figure 13 shows the elongation at break of the original and recycled epoxy vitrimers. Like the tensile strength, the elongation at break of the original epoxy vitrimers decreases with the epoxy/acid ratio. However, different from the tensile strength, the recycling lowers the elongation at break of the original epoxy vitrimers. A similar trend was reported in the tetrafunctional epoxy/dimerized acid vitrimers [44]. At the same recycling time, the elongation at break of recycled epoxy vitrimers also decreases with the epoxy/acid ratio.

Tensile toughness, the area under the stress–strain curve, indicates the energy absorption up to the material’s failure [53,54,55]. The tensile toughness of the original and recycled epoxy vitrimers is shown in Figure 14. Similar to the tensile strength and elongation at break, the tensile toughness of the original epoxy vitrimers decreases with the epoxy/acid ratio. Except for the first recycling of EV50 and EV55, the recycling increases the tensile toughness of the original epoxy vitrimers due to the increase in both the tensile strength and elongation at break. The tensile toughness of the recycled epoxy vitrimers increases with the recycling time, except for the fourth recycling of EV45.

## 4. Conclusions

This work investigated the effect of epoxy/acid stoichiometry on the cure behavior and mechanical properties of epoxy vitrimers. Epoxy vitrimers with three epoxy/acid stoichiometries from the deficit to the excess of epoxy oligomers were prepared and recycled. Because of the catalytic ring-opening polymerization at elevated temperature as well as the steric effect of long-chain fatty acids, the epoxy vitrimer with the excess of epoxy oligomers shows the lowest reaction rate and conversion during the isothermal curing. The swelling ratio of the original epoxy oligomers decreases with the epoxy/acid ratio, whereas the gel content shows a contrary trend. The recycling decreases the swelling ratio but increases the gel content in the original epoxy vitrimers. The swelling ratio of the recycled epoxy vitrimers decreases with the recycling time, while the gel content shows an opposite trend. For the epoxy vitrimer with an excess of epoxy oligomers, the recycling increases the storage modulus. The T_g_ of the original epoxy vitrimers decreases with the epoxy/acid ratio. The recycling decreases the T_g_ of the original epoxy vitrimer with the deficit epoxy oligomers. However, the T_g_ of the original epoxy vitrimer with the excess of epoxy oligomers increases after the recycling. In addition, the T_g_ of the recycled epoxy vitrimer with an excess of epoxy oligomers increases with the recycling time. Both the tensile strength and toughness of the original vitrimers increase with the increase in the epoxy/acid ratio and increase after the recycling. However, the elongation at break shows a contrary trend. Both the tensile strength and toughness of the recycled epoxy vitrimers with the stoichiometry and the excess of epoxy oligomers increase with the recycling time.

## Figures and Tables

**Figure 1 molecules-27-06335-f001:**
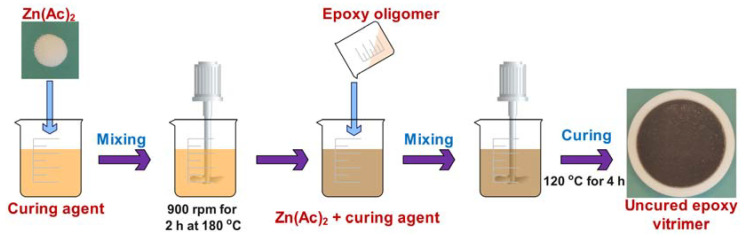
Schematic presentation for the preparation of epoxy vitrimer.

**Figure 2 molecules-27-06335-f002:**
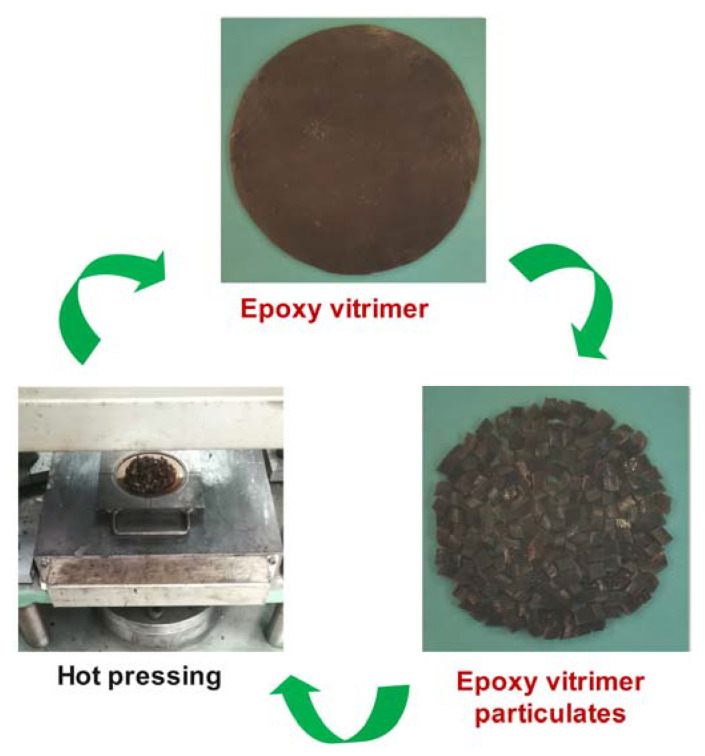
Schematic illustration for the recycling of vitrimer.

**Figure 3 molecules-27-06335-f003:**
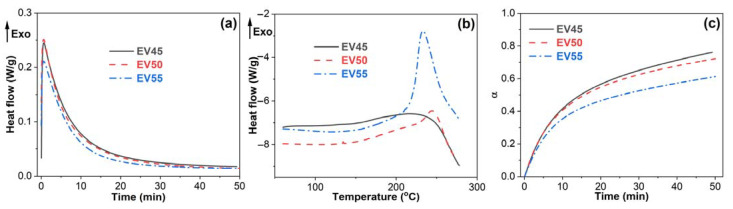
(**a**) Isothermal curing at 120 °C, (**b**) dynamic curing and (**c**) extent of reaction as a function of time at 120 °C of the original epoxy vitrimers.

**Figure 4 molecules-27-06335-f004:**
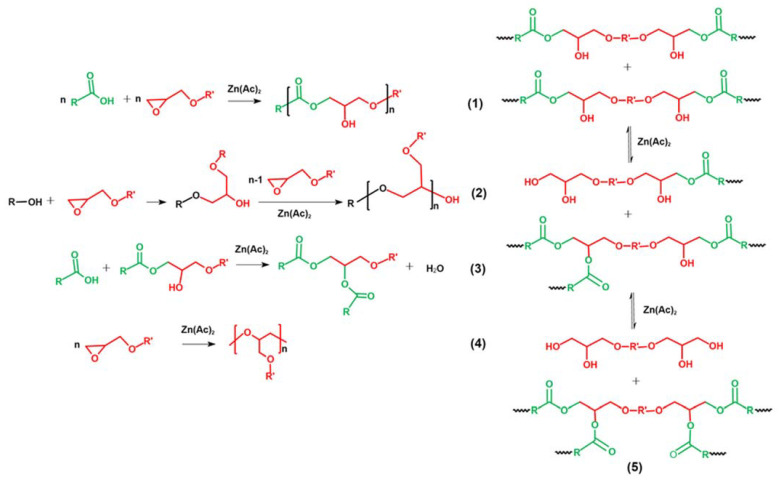
Main reactions in the cure reaction of epoxy vitrimers.

**Figure 5 molecules-27-06335-f005:**
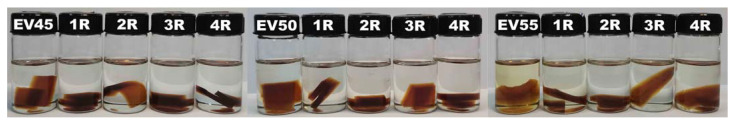
Solvent stability of the original and recycled epoxy vitrimers in toluene at room temperature after 48 h.

**Figure 6 molecules-27-06335-f006:**
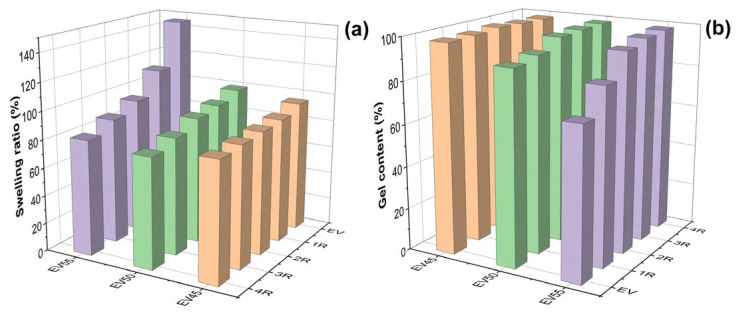
(**a**) Swelling ratios and (**b**) gel contents of the original and recycled epoxy vitrimers after immersing in toluene for 48 h at room temperature.

**Figure 7 molecules-27-06335-f007:**
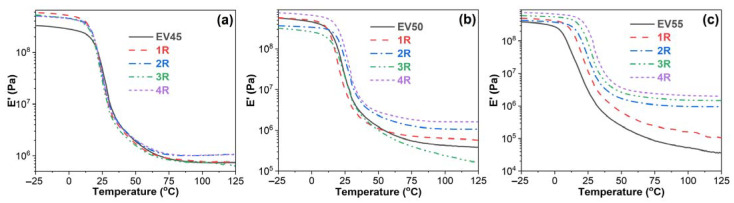
Storage modulus versus temperature curves of the original and recycled epoxy vitrimers: (**a**) EV45, (**b**) EV50 and (**c**) EV55.

**Figure 8 molecules-27-06335-f008:**
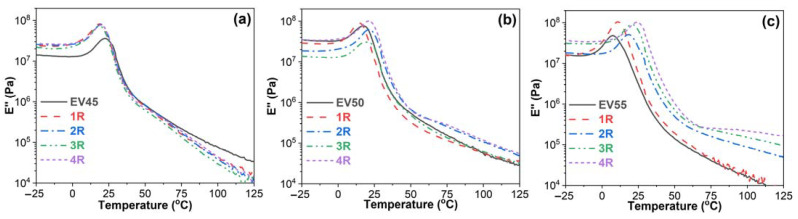
Loss modulus versus temperature curves of the original and recycled epoxy vitrimers: (**a**) EV45, (**b**) EV50 and (**c**) EV55.

**Figure 9 molecules-27-06335-f009:**
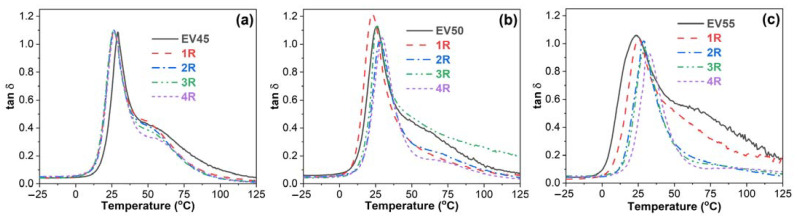
Damping factor versus temperature curves of the original and recycled epoxy vitrimers: (**a**) EV45, (**b**) EV50 and (**c**) EV55.

**Figure 10 molecules-27-06335-f010:**
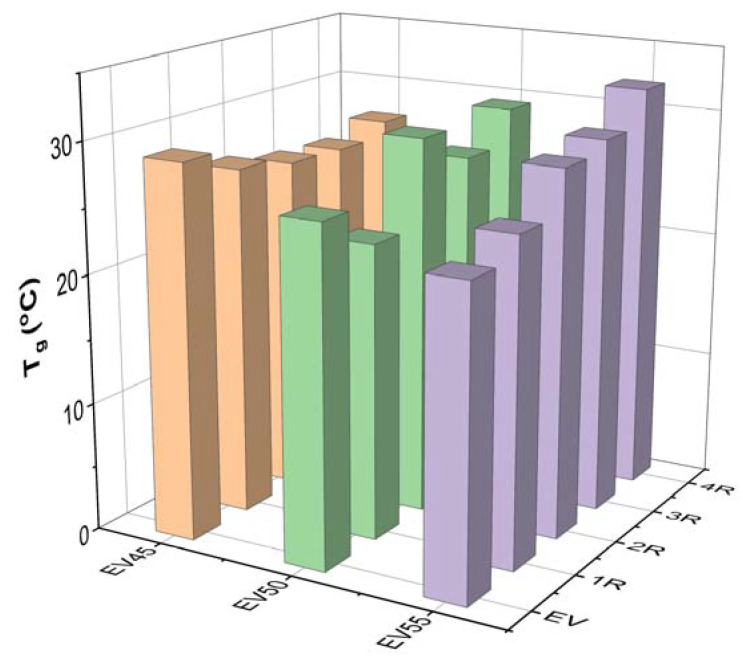
T_g_s of the original and recycled epoxy vitrimers.

**Figure 11 molecules-27-06335-f011:**
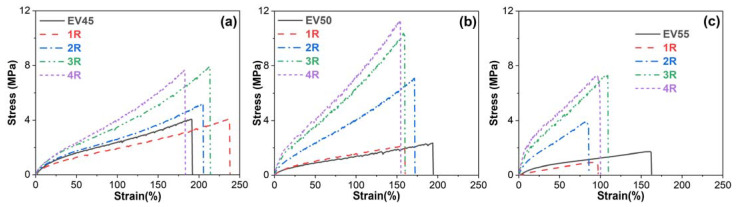
Stress–strain curves of the original and recycled epoxy vitrimers: (**a**) EV45, (**b**) EV50 and (**c**) EV55.

**Figure 12 molecules-27-06335-f012:**
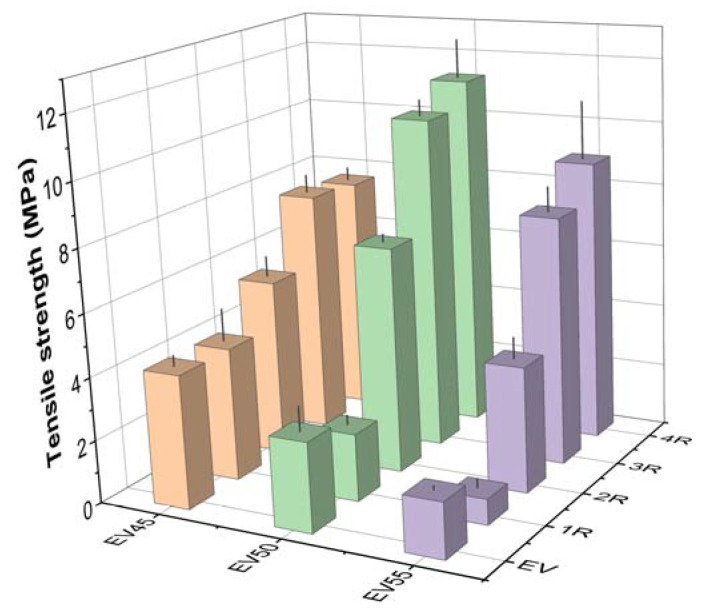
Tensile strength of the original and recycled epoxy vitrimers.

**Figure 13 molecules-27-06335-f013:**
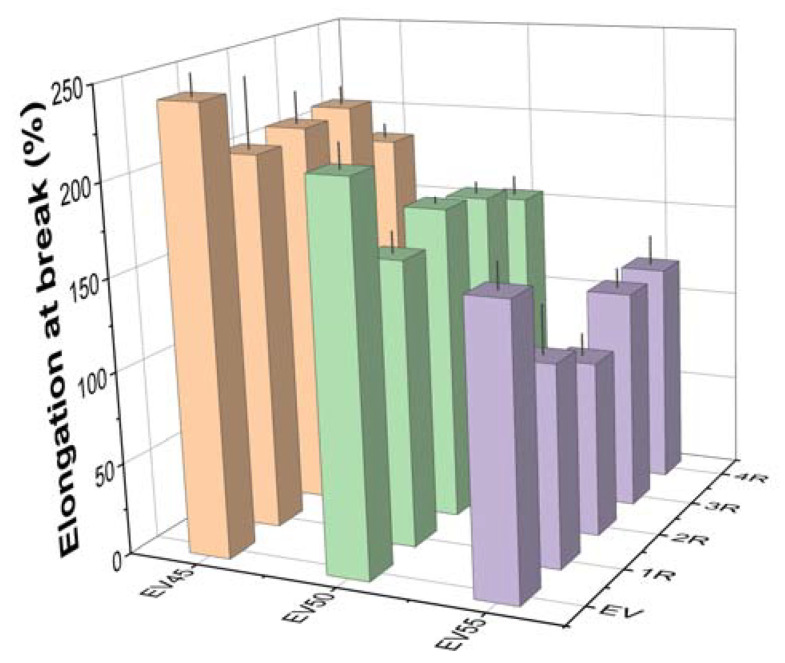
Elongation at break of the original and recycled epoxy vitrimers.

**Figure 14 molecules-27-06335-f014:**
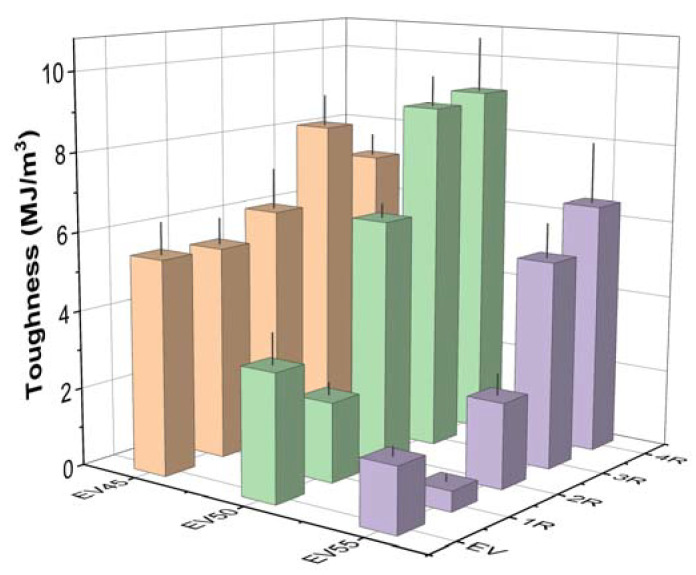
Tensile toughness of the original and recycled epoxy vitrimers.

**Table 1 molecules-27-06335-t001:** Chemical structures and molecular weights (MWs) of reagents.

Reagent	Structure	MW (g/mol)
Epoxy oligomer	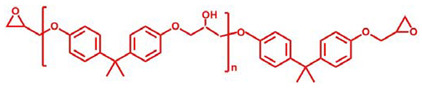	192
Mixture of fatty acid	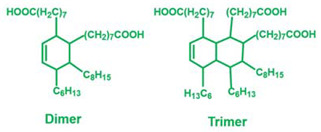	296
Zinc acetate	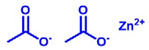	183.48

## Data Availability

All data are available in the manuscript.

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
