# Peer review of "Influence of the Epoxy/Acid Stoichiometry on the Cure Behavior and Mechanical Properties of Epoxy Vitrimers"

_molecules, 2022, doi:10.3390/molecules27196335_

Round 1

Reviewer 1 Report

The current paper on theInfluence of the Epoxy/Acid Stoichiometry on the Cure Behavior and Mechanical Properties of Epoxy Vitrimers” is a good work on vitrimers. However, I thought that the novelty of work is not enough for publication in Molecules. The work is not innovative in terms of the type of materials or experimental conditions. The obtained results of the mechanical properties are also much weaker than usual epoxies. (Figure 8 is obviously wrong and the failure strain results should be replaced). So, I recommend this paper to be rejected.

Reviewer 2 Report

In the reviewer’s opinion, there are some new technical contributions reported in this manuscript. The authors are requested to respond to the following comments before final recommendation:

11. Section 2 and Section 3 need to be swapped. The “Materials and Methods” need to be presented first before “Results and Discussion”.

22. Change “Scheme 1” to “Figure 2”. Accordingly, please update the figure caption number for all subsequent figures.

33. The y-axis of Figure 8 is wrong. It should be “elongation at break”. Please correct it.

44. It is mandatory to show all force-displacement or stress-strain curves for the tensile test, and provide brief explanations on how the tensile strength and elongation at break are identified from the curves.

Round 2

Reviewer 1 Report

The current paper on the “Influence of the Epoxy/Acid Stoichiometry on the Cure Behavior and Mechanical Properties of Epoxy Vitrimers” is a good work on vitrimers. However, I still thought that the novelty of work is low. The effect of stoichiometry on thermo-mechanical properties of epoxy vitrimers was studied, before. The author also should respond to the following notes:  

Page 5: “…the conversions of EV45, EV50 and EV55 are 0.76, 0.72 and 0.61, respectively, indicating

that the conversion and reaction rate of the epoxy vitrimers decreases with the epoxy/acid 180 ratio”. Why? Why the conversion was not completed at equal stoichiometry (acid/epoxy) ?

Figure 7.b. EV50 after recycling had lower storage modulus if glassy state (2R and 3R) and rubbery state (3R). What was the reason? The author should completely discus about that.

Page 8: “…  it is interesting to note that another broad glass transition (secondary transition 263

or beta transition) appears between 40 oC and 100 oC …“. Why the author thinks that is beta transition? It can be another alfa transition?  Tan(delta) is not enough for this conclusion. Please add the loss modulus data.

Page 9: “The Tg of the original epoxy vitrimers decreases with the epoxy/acid ratio.” Why? It was expected to have an optimum point at the same stoichiometric ratio. Why EV55 showed opposite trend with recycling? It is needed to more discussion.

Is there any correlation between tensile strength or modulus and glass transition of conversion?

Reviewer 2 Report

In the reviewer's opinion, the authors have responded all comments satisfactorily. Hence, the reviewer recommends this manuscript to be accepted for publication.

Author Response

Thank you.